# One-Shot Preparation of Polybasic Ternary Hybrid Cryogels Consisting of Halloysite Nanotubes and Tertiary Amine Functional Groups: An Efficient and Convenient Way by Freezing-Induced Gelation

**DOI:** 10.3390/gels7010016

**Published:** 2021-02-05

**Authors:** Nur Sena Okten Besli, Nermin Orakdogen

**Affiliations:** 1Department of Civil Engineering, Istanbul Kultur University, Bakırkoy, 34158 Istanbul, Turkey; nursenaokten@gmail.com; 2Department of Chemistry, Soft Materials Research Laboratory, Istanbul Technical University, Maslak, 34469 Istanbul, Turkey

**Keywords:** hybrid, halloysite, cryogel, smart-responsive, elasticity, swelling

## Abstract

A convenient method for the preparation of polybasic ternary hybrid cryogels consisting of Halloysite nanotubes (HNTs) and tertiary amine functional groups by freezing-induced gelation is proposed. Ternary hybrid gels were produced via one-shot radical terpolymerization of 2-hydroxyethyl methacrylate (HEMA), 2-acrylamido-2-methyl-1-propane sulfonic acid (AMPS), and DEAEMA in the presence of HNTs. The equilibrium swelling in various swelling media and the mechanical properties of the produced ternary hybrid gels were analyzed to investigate their network structure and determine their final performance. The swelling ratio of HNT-free gels was significantly higher than the ternary hybrid gels composed of high amount of HNTs. The addition of HNTs to terpolymer network did not suppress pH- and temperature-sensitive behavior. While DEAEMA groups were effective for pH-sensitive swelling, it was determined that both HEMA and DEAEMA groups were effective in temperature-sensitive swelling. Ternary hybrid gels simultaneously demonstrated both negative and positive temperature-responsive swelling behavior. The swelling ratio changed considerably according to swelling temperature. Both DEAEMA and HEMA monomers in terpolymer structure were dominant in temperature-sensitive swelling. Mechanical tests in compression of both as-prepared and swollen-state demonstrated that strength and modulus of hybrid cryogels significantly increased with addition of HNTs without significant loss of mechanical strength. Ultimately, the results of the current system can benefit characterization with analysis tools for the application of innovative materials.

## 1. Introduction

With the rapid development of nanotechnology, novel nanoparticles have been used for manufacturing advanced nanomaterials, showing unique design possibilities and unusual property combinations [1,2,3,4,5]. By virtue of their large surface-to-volume ratio, which increases the number of interactions between the nanoparticles and polymer-based matrix, the nanomaterials may result in advanced structures for applications involving controlled/rapid release of drugs/solutes, active food packaging, chemical reactions, and higher barrier performance. Nanocomposite design offers the advantage of achieving favorable elasticity through the incorporated nanosized reinforcement components in the form of layered nanoparticles, fibers, whiskers, and nanotubes [6,7,8,9,10]. In the present study, polybasic ternary hybrid gels consisting of Halloysite nanotubes (HNTs) and tertiary amine functional groups were designed with stimuli-responsive nature. The design of smart hybrid materials with synergistically enhanced strength and combined responsive properties is presented in an effective and convenient way.

Much effort currently focuses on HNT-reinforced polymeric material development through nanotechnology for specialty use. From aluminosilicate nanoclay mineral family, HNT has a hollow nanotubular structure formed by wrapping of the aluminosilicate layers roll into a cylinder. Naturally-occurring HNT belongs to dioctahedral 1:1 clay mineral of the kaolin group with a predominantly hollow tubular structure with chemical formula Al_2_Si_2_O_5_(OH)_4_·nH_2_O, where n = 0 for halloysite-(7 Å) and n = 2 for halloysite-(10 Å). The length of tube-shaped structures ranges from 150 nm to 2 µm; the outer diameter varies between 20 and 100 nm; and the lumen diameter ranges 5–30 nm. The tubular shape of HNT contains two layers: tetra- and octahedral layers. There are two types of hydroxyl groups present in the structure of HNTs: the inner group is located between nanotubes layers and the second group is a surface hydroxyl group. Therefore, the internal surface is positively charged and consists of gibbsite octahedral sheet (Al-OH) groups, while its outer surface is negatively charged and composed of siloxane groups (Si-O-Si). Compared to kaolin, the unit layers are separated by a layer of water molecules [11]. The empty space inside the tubular structure enables the entrapment of hydrophilic and lipophilic substances as well as the design of responsive nanocomposites with synergistically improved adhesivity and strength, which are needed for biomedical applications [12,13]. Luo and Mills developed stable drug delivery chitosan/HNT nanocomposite hydrogels and used HNTs as a bulk filler to improve the mechanical weakness of chitosan-based hydrogels, which are widely used in biomedical applications due to their ability to mimic the extracellular matrix of many tissues. The resultant gentamicin-doped nanocomposite hydrogels provided a sustained pattern of drug release and inhibited bacterial growth [14]. Kennouche et al. prepared ternary poly(3-hydroxybutyrate-co-3-hydroxyvalerate)/poly(butylene succinate)/halloysite nanocomposites by melt compounding. Incorporation of HNT significantly improved the fire reaction, while their thermal stability decreased as a result of typical co-continuous and nodular morphology of the nanocomposites [15]. Using green chemistry, Pavlinakova et al. prepared nanofibrous elastic materials based on hydrophobic poly(ε-caprolactone) and hydrophilic gelatin reinforced with HNTs by electrospinning process Three different kinds of HNTs with similar aspect ratio but different length and inner diameter were incorporated to test reinforcing effect of each type of HNTs. The highest improvement of mechanical properties was obtained with the addition of 0.5 wt% of HNTs [16].

Various approaches to improve the mechanical weakness of nano (hybrid) materials are reported in the literature [17,18,19,20]; among them, the most common procedure is to carry out the polymerization at a temperature below the freezing point of polymerization solvent [21,22,23]. Cryogelation is a simple method in which the gelation process occurs under semi-frozen conditions leading to a polymer network cross-linked around ice crystals. As cooling commences, the formation of ice crystals (in the case of aqueous systems) starts upon freezing. The pre-gel reaction mixture containing gel-forming agents is concentrated in unfrozen liquid microphases, i.e., cryoconcentration takes place along with the ice crystals which act as pore-forming agents. In fact, the cryoconcentration of pre-gel reaction mixture promotes the gel-formation either by accelerating the rate of gelation or by lowering the critical concentration required for gelling. An interconnected porous cryogel network is formed by thawing of the ice crystals. The size and shape of these pores mainly depend on the type and concentration of monomers in the pre-gel reaction mixture, the rate of solvent crystallization, efficiency and type of crosslinking, efficient cryoconcentration, the temperature and duration of cryogelation, and cooling rate [24,25,26,27]. Recently, Stoporev et al. reported an interesting application of HNTs containing nanocomposite cryogels and showed that a cryogel matrix composed of poly(vinyl alcohol) (PVA) and natural HNTs can be used for the cold storage. The clay nanotubes promoted the hydrate nucleation, reducing the supercooling of the formed hydrate, while PVA provided the formation of a cryogel and the sedimentation stability of HNT inside the dispersion [28].

Research on poly(alkyl methacrylate)-based hybrid cryogels and hydrogels containing HNTs has apparently not been conducted to provide proof of the full effect of HNTs as a reinforcing agent since the attention for the preparation of hybrid cryogels is relatively new to some extent. The purpose of this research was to develop a synthetic way to prepare nanostructured ternary systems with typical pH- and temperature-sensitive alkylmethacrylate-based polymer properties, but with the enhanced flexibility and thermal properties of HNTs depending on the network topology. Another key focus of this study was to assess the physical phenomena that influence the mechanical properties of terpolymer–HNT hybrid materials. By demonstrating the importance of monomer(s) selection and terpolymer composition, this work presents new insight into the area of excipient design of hybrid materials to fine-tune the mechanical properties and maintain desired swelling extent of poly(alkyl methacrylate)-based hybrid gels. The development of one-shot preparation of polybasic ternary hybrid cryogels containing nanoparticle HNT, neutral monomer 2-hydroxyethyl methacrylate (HEMA), anionic monomer 2-acrylamido-2-methyl-1-propane sulfonic acid (AMPS), and cationic monomer (DEAEMA) in the presence of chemical crosslinker diethyleneglycol dimethacrylate (DEGDMA) is presented. The efforts to evaluate the structure–property relationships may help to understand the use of HNT as a nanocontainer for encapsulation of various functional agents as well as for producing hybrid materials by freezing-induced gelation.

## 2. Results and Discussion

Ternary hybrid networks with variable halloysite content, from 0% to 3.20% (*w*/*v*), were prepared using the solvent casting method under cryogenic conditions. The swelling response, mechanical behavior, and thermal properties of the prepared ternary hybrid gels were investigated to estimate their suitability as multi-responsive materials. The results show that this work opens a new pathway to fabricate ternary hybrid systems containing nanotubular structures and tertiary amine-functional groups.

### 2.1. Ternary Hybrid Gel Preparation

Halloysite/ternary hybrid networks in the form of terpolymer hydrogels and cryogels with different nanotube contents were prepared via the free-radical (cryo)polymerization of neutral monomer HEMA, anionic comonomer AMPS, and weakly basic termonomer DEAEMA. The efficient and convenient method of freezing-induced gelation provided a facile method to obtain ternary hybrid cryogels. Diethyleneglycol dimethacrylate (DEGDMA) was used as the chemical crosslinking agent and the crosslinker ratio (mole ratio of the crosslinker DEGDMA to the monomers HEMA, AMPS, and DEAEMA) was fixed at 1/90. To initiate the reactions, 2.63 mM of initiator APS and 24.9 mM (0.375 *v/v* %) of catalyst TEMED were used. Table 1 presents the sample codes and the content of the ternary hybrids. The mole ratio of monomers HEMA, AMPS and DEAEMA was fixed at 86:4:10 mol%, and the HNT content of hybrid gels (C_HNT_ = % *w/v* in the reaction solution) was varied from 0% to 3.20% (*w*/*v*) with respect to the monomers. In the pre-gel solution, the total molar concentration C_0_ of the monomers was fixed at 26.6% (*w*/*v*). Two different polymerization temperatures were selected for the production of ternary hybrid materials in two different network structures: ternary hybrid cryogels (Cgs) and hydrogels (Hgs) were prepared and are referred to as HNTm/PHAD-Cgs and Hgs, with H, A, and D representing the monomers HEMA, AMPS, and DEAEMA, respectively, and m referring to HNT concentration C_HNT_ = % (*w*/*v*) in the reaction solution, which ranged 0.30–3.20% (*w*/*v*), as given in Table 1. Note that the gel given with the code HNT0/PHAD in Table 1 refers to the HNT-free terpolymer gel with the structure of poly(HEMA-co-AMPS-co-DEAEMA), henceforth designated as PHAD gel.

Firstly, the specified amount of HNT was added to 6.0 mL of deionized water and magnetically stirred at room temperature for 4 h. Then, 1.720 mL of HEMA, 580 μL of AMPS stock solution (5 g AMPS/25 mL water), and 285 μL of DEAEMA monomer were added sequentially, and, after each monomer addition, the pre-solution was stirred for 1 h. After an average mixing time of 7 h, 1.0 mL of stock solution of catalyst TEMED (0.375 mL TEMED/10 mL water) and 40 μL of crosslinker DEGDMA (11.5 mmol) were added sequentially and mixed again for 30 min in between. Finally, 1.0 mL stock solution of APS (0.08 g APS/10 mL water) was included, and the pre-gel solution was immediately stirred at 2000 rpm for 30 s. The final mixture was slowly stirred and transferred into the several propylene syringes. To obtain ternary hybrid cryogels, henceforth designated as HNTm/PHAD-Cgs, the freezing induced gelation via cryopolymerization was conducted at −18 °C for 48 h and the cryogelation temperature was lowered with a slow cooling rate, 5 °C min^−1^. To make comparisons, ternary hybrid hydrogels, henceforth designated as HNTm/PHAD-Hgs, were prepared via conducting the conventional free radical crosslinking polymerization reaction at 24 °C for 48 h. A diagram illustrating the structure and preparation process of ternary hybrid HNTm/PHAD gels is presented in Scheme 1.

### 2.2. Structural Properties of Ternary Hybrid Networks

ATR-FTIR was used to characterize the structure of as-prepared ternary hybrid gels. In the structure of HNT, two different hydroxyl groups, inner and outer hydroxyl groups, are located on the surface and between the layers of the nanotube, respectively. The internal surface of HNT is covered by aluminol groups and the external surface is predominantly composed of siloxane (Si-O-Si) groups, with some silanol (Si-OH) and aluminol (Al-OH) groups exposed at the edges of the tube [29,30].

In Figure 1, the absorption bands around 908 and 1018 cm^−1^ in ATR-FTIR spectra of pristine HNT were related to the Al-OH vibrations and Si-O stretching. Upon polymerization, the band at 908 cm^−1^ was shifted to 911 cm^−1^, whereas the band at 1018 cm^−1^ was shifted to 1025 and 1027 cm^−1^ for HNT10/PHAD and HNT11/PHAD Hgs, respectively. The observed shift corresponded to the interaction of OH groups inside the lumen of HNT that indicated the penetration of terpolymer PHAD chains to some extent into the nanotubes. The peaks at 747 and 796 cm^−1^ were assigned to the perpendicular and symmetric stretching of Si-O, while the peak at 1120 cm^−1^ was due to in plane Si-O stretching. This peak shifted slightly to 1159 cm^−1^ after polymerization. In the spectra of hybrid gels, the band at 1060 cm^−1^ was due to the formation of Si-O-Si groups and indicated the interaction between the surface of the HNT and terpolymer PHAD. The peaks at 3626 and 3696 cm^−1^ were due to O-H stretching of inner hydroxyl groups and O-H stretching of inner-surface hydroxyl groups, respectively. The presence of OH vibration bands at around 3625 and 3699 cm^−1^ showed that hydrogen bonds were not formed between HNT and PHAD chains as the OH groups of HNT were not easily accessible. The presence of respective hydroxyl group and corresponding absorption of moisture was detected at 3416 cm^−1^. The peak at 1637 cm^−1^ was attributed to O-H deformation of interlayer water, while the typical stretch vibration band of the ester carbonyl groups of terpolymer PHAD at 1640 cm^−1^ overlapped with the OH deformation band of water. In the spectrum of HNT-free terpolymer gels, HNT0/PHAD, and terpolymer hybrid gels, HNT10/PHAD and HNT11/PHAD, the −C=O stretching bands due to ester carbonyl group were seen at 1717/1718 cm^−1^. A wide band detected between 3700 and 3050 cm^−1^ with a maximum at 3367 cm^−1^ was assigned to the -OH stretching vibrations. The characteristic absorption bands at 2945 and 2949 cm^−1^ were attributed to C-H stretching vibrations for methyl and methylene groups from aliphatic groups. The peaks at 848 and 892 cm^−1^ were assigned to =C-H stretching.

The absorption band at 1179 cm^−1^ were assigned to the symmetric stretching vibration of C-N band associated with the DEAEMA units, while the contribution of the ester group of DEAEMA was observed as a shoulder at 1182 cm^−1^. The absorption bands at 2871–2877 cm^−1^ were the bending vibrations of −CH_2_ and −CH groups of the −N(C_2_H_5_)_2_ groups, while the bands at 1580 cm^−1^ were observed due to the coordination of the free electron pair of the tertiary amino group with the carboxyl from DEAEMA. The distortion vibration bands of methylene were observed at around 1021 cm^−1^. The peak at 1640 cm^−1^ corresponded to the carboxylate group of AMPS, and the carboxylate group symmetric stretching vibrations were indicated by a band detected at 1449–1451 cm^−1^. The S=O asymmetric stretching of AMPS units occurred at 1386 cm^−1^, while the C-S stretching of AMPS occurred at 660 cm^−1^. The asymmetric sulfonic acid (SO_2_) stretching was observed at around 1252 cm^−1^, while symmetric stretching vibration was observed at 1079 cm^−1^.

### 2.3. Thermal Stability of Ternary Hybrid Gels

To determine the thermal stability of terpolymer hybrid materials as well as their compositional properties, Figure 2A shows the plot of M(T)/M_0_ percent, the fraction of the sample mass remaining after heating, where M_0_ refers to initial mass and M(T) refers to the mass at temperature T. The derivative weight loss (%/°C) curves of pristine HNT, HNT-free hydrogel and cryogel (HNT0/PHAD-Hgs and -Cgs), and ternary hybrid hydrogel and cryogel (HNT10/PHAD and HNT11/PHAD-Hgs and -Cgs) samples prepared at HNT content 2.80% and 3.20% (*w*/*v*) are shown in Figure 2B. In Table 2, the maximum degradation temperatures (T_max_) and char residues at 700 °C are presented. Upon heating, HNT-10 Å undergoes dehydration attributed to the presence of interlayer water. As shown in Figure 2C, the thermal reaction of HNT can be divided into mainly three steps: (i) Dehydration and the slight weight loss below 150 °C (low-temperature reaction) is due to the evaporation of the adsorbed water on the surface of HNTs. (ii) Dehydroxylation between 350 and 550 °C (intermediate-temperature reaction is due to the condensation reaction of Al-OH and Si-OH groups. The release of OH groups from the octahedral coordinated Al^3+^ ion results in the formation of the “metahalloysite”. (iii) High temperature reactions above 600 °C result in the formation of a distinct alumina rich phase and amorphous SiO_2_. The residue amount at 800 °C was 73.5%. In the case of HNT-free hydrogel and cryogel (HNT0/PHAD-Hgs and -Cgs), the degradation process occurred in two stages, with the initial stage being attributed to the loss of bound water. The first degradation stage occurred in the range of 285–360 °C with a weight loss of approximately 17.7% and 19.9% for HNT0/PHAD-Hgs and -Cgs, respectively. The second stage ranged from 360 to 495 °C with the weight loss of 39.2% and 32.8% attributed to the destruction of whole terpolymer network. HNT-containing ternary hybrid gels started to degrade at higher temperatures when compared to the HNT-free gels. By incorporation of HNT into terpolymer matrix, the hybrid gels exhibited two-step degradation profile, while HNT11/PHAD-Cgs sample exhibited one-step degradation profile that reflected a single peak in the DTG curve. A significant weight loss was obtained between 360 and 500 °C corresponding to terpolymer network degradation. For HNT10/PHAD-Hgs sample containing 2.80% (*w*/*v*) HNT, the two-stage weight loss is particularly noteworthy.

After the first stage loss ascribed to the vaporization of bound water, the second weight loss at 275–360 °C was due to the decomposition and fracture of the pendant functional groups including diethylamino ethyl groups, sulfate, hydroxyl-, and carboxyl- groups of DEAEMA, HEMA, and AMPS and the crosslinker DEGDMA on the terpolymer matrix. The second weight loss at about 360–500 °C was primarily attributed to terpolymer degradation and random chain scission. The weight loss trend was similar to those for HNT11/PHAD-Hgs sample containing 3.20% (*w*/*v*) of HNT. An increase in the amount of pristine HNT accelerated the decomposition in the first stage while slightly increased Tmax_2_ of the hybrid gels in the second stage compared to the neat terpolymer matrix as well as improved the thermal stability of hybrid gels as seen from char residue at 700 °C. This result may be attributed to two opposite effects of pristine HNTs. Due to the presence of hydroxyl groups (Al-OH) acting as Brönsted active sites, HNT displays a catalytic effect [31]. Besides, due to the barrier and entrapment effects of the nanotubes, HNTs can capture the volatile products into their lumen and thus prevent their escape in the degradation process, which enhances the thermal stability of hybrid gels [32]. Thus, DSC analyses were performed over the temperature range from −20 to 200 °C to verify the hybrid structures. In Figure 3, the DSC endothermic curves for HNT-free, HNT0/PHAD, and ternary hybrid HNTm/PHAD-Hgs and -Cgs are displayed and the onset temperature (T_onset_), glass transition temperature (T_g_), and heat capacity change (Δ*C_p_*) values of the samples are summarized in Table 3. Using the second heating curve, T_g_ determined by the midpoint of the transition region for terpolymer hybrid gels HNT0/PHAD to HNT11/PHAD varied as a function of respective HNT content. The observation of a single T_g_ for HNT-free gels as well as for HNT-containing hybrid gels supported the compatibility and the uniform distribution of HNT and monomers—DEAEMA, HEMA, and AMPS—among the terpolymer chain. As the single peak of T_g_ usually reflects the homogeneous distribution of components, this result indicates that the random terpolymer network of P(HEMA-AMPS-DEAEMA) containing various amount of HNT was successfully prepared. It can be clearly seen that the T_g_ values of HNT-containing hybrid gels were greater than those of HNT-free terpolymer gels due to limited movement of the terpolymer chains in the presence of nanotubes leading to an increase in T_g_ of ternary hybrid gels. Further, Δ*C_p_* values of the ternary hybrid gels also decreased with increasing HNT content. Obviously, incorporation of HNT into terpolymer PHAD structure appeared to increased crosslink density and thus slightly increased T_g_ values of PHAD terpolymer networks. This increase can be attributed to the interaction between the terpolymer PHAD network and the HNTs that limit the mobility of the terpolymer PHAD chains by the dispersed nanotubes.

### 2.4. XRD Measurements of Ternary Hybrid Gels

Figure 4 shows the XRD pattern of pristine HNT, HNT-free hydrogel and cryogel (HNT0/PHAD-Hgs and -Cgs), and ternary hybrid hydrogel and cryogel (HNT2/PHAD and HNT10/PHAD-Hgs and -Cgs) samples containing 0.50% and 2.80% (*w*/*v*) of HNT. The XRD pattern of pristine HNT reflected a monoclinic-domatic crystallographic character of halloysite-(10 Å). The characteristic sharp peak at 8.86° (2θ) of HNT corresponding to d001 basal spacing of 9.966 Å, determined by using Bragg’s law, occurred due to HNT multiwall reflection. This result indicates that HNT was mainly in the hydrated form typically referred to as 10 Å-halloysite.

Two other peaks of HNT detected at 2θ of 20.10° and 24.58° indicative of the tubular halloysite structure were confirmed with the presence of d100 and d002 basal reflection, which is equivalent to basal spacing of 4.412 and 3.617 Å, respectively. The 24.58° peak was due to the possible presence of nacrite (N) and kaolinite (K), together, while the peaks around 35.23° were attributed to the presence of minerals—magnetite (M), dictite (Di), and hematite (Hi). The XRD pattern of HNT-free hydrogel and cryogel (HNT0/PHAD-Hgs and -Cgs) showed a wide reflection centered ~17.42° (2θ), (d = 5.083 Å), which was attributed to the amorphous structure of terpolymer PHAD. The patterns of HNT2/PHAD-Hgs and -Cgs containing 0.50% (*w*/*v*) of HNT showed broad diffraction between 2θ = 10.65° and 21.60° with two maxima at around 12.25° and 17.62° owing to dehydroxylation and the formation of an X-ray amorphous product, metahalloysit. By terpolymerization, HNT did not lose its characteristic peaks in the corresponding 2θ region. The characteristic peak appeared only as a small shoulder for hybrid hydrogels loaded with 0.50% (*w*/*v*) of HNT; however, it became more visible, especially by the increasing loading amount up to 2.80% (*w*/*v*) of HNT in the pattern of HNT10/PHAD. Upon increased addition of HNT, the characteristic peak appeared at 11.64° with d001 basal spacing of 7.592 Å. This showed that, due to the terpolymerization, PHAD terpolymer chains are placed on the surface of the HNT nanotubes. There were slight changes in the d020, d110, and d002 basal spacing of 4.390 and 3.574 Å. Such spacing was shifted to a lower 2θ equal to about 20.20° and 24.88°, respectively. This result indicates that the crystal structure of HNT remained unchanged with the formation of the ternary hybrid network [33].

### 2.5. Mechanical Analysis of Ternary Hybrid Gels

Uniaxial compressive elasticity was utilized to measure the maximum stress that the terpolymer HNTm/PHAD gels would sustain over time. The compression modulus of each terpolymer network was determined from the slope of the linear portions of the compression stress–strain curves, as presented in Figure 5. Optical images of HNT9-PHAD-Hgs and HNT9-PHAD-Cgs containing 2.50% (*w*/*v*) HNT in Figure 5A indicate that the samples are highly elastic to withstand high level of deformation under finger compression. The stress–strain curves showed that all the prepared ternary hybrid samples were elastic, since the strain increased linearly with the compressive stress. Moreover, it was observed that the strain was much higher for the ternary hybrid gels containing 5.20% (*w*/*v*) of HNTs in comparison with the other samples. When comparing Figure 5B1,B2, it is clearly observed that the slope of the stress–strain curves of HNTm-PHAD-Hgs decreases after the swelling process. This is the expected result of the swelling process according to rubber elasticity theory. However, compared to HNTm-PHAD-Hgs, in Figure 5B2, an obvious increase in the stress at all strains was detected for HNTm-PHAD-Cgs, as shown in Figure 5C. The observed increase here for the ternary hybrid cryogels is the result of the morphological changes caused by the cryogelation process in the gel structure.

According to Equation (3), at low strains, the slope of the straight lines obtained from the plot of compression stress σ versus $-(\alpha -\alpha^{-2})$ gives the compressive elastic modulus of ternary HNTm/PHAD gels in either as-prepared state *G*_0_ or swollen state *G*. From these slopes, the compression moduli of ternary HNTm/PHAD gels determined in as-prepared state or swollen state in water are compared in Figure 6 against HNT content in the feed. It can be clearly seen that the amount of inorganic nanotubes deeply influences the compressive elasticity of ternary hybrid gels. For HNTm/PHAD-Hgs, the compressive elastic moduli of as-prepared state and swollen state both significantly increased with an increase in HNT loading from 0% to 0.70% (*w/v*), as compared with that of HNT-free terpolymer network. In the as-prepared state, the compressive modulus increased sharply by 2.8 folds from 1.36 ± 0.07 Pa to 3.85 ± 0.8 kPa (HNT3/PHAD-Hgs) and by 4.9 folds to 6.64 ± 1.2 kPa (HNT11/PHAD-Hgs) with an increase in HNTs loading from 0% to 0.70% and 3.20% (*w/v*), respectively. After equilibrium swelling, the elastic modulus of ternary hybrid cryogels was found to be higher at all HNT concentrations than that of hydrogels. The swollen compressive moduli increased from 1.11 ± 0.4 to 3.11 ± 0.6 kPa for HNTm/PHAD-Hgs, while a sharp increase from 2.75 ± 0.5 to 8.74 ± 0.7 kPa was obtained for HNTm/PHAD-Cgs with an increase in HNTs loading from 0% to 3.20% (*w/v*). This improvement indicated that HNTs acted as a reinforcing agent in the ternary network due to uniform dispersion and to the strong interfacial interactions between hydroxyl, oxygen atoms, and CO groups of ternary network with the external surface siloxane (Si-O-Si) groups of HNTs as well as due to the hydrogen bonding interactions between the amine and oxygen atoms of network chains and the interlayer inner surface Al-OH groups. Since the DEAEMA monomer in the ternary network structure is a cationic monomer, when considered in terms of similar interactions, the cationic chitosan monomer can be given as an example of monomers with similar structure. The hydroxyl groups and amine groups on chitosan can interact with the Si-O bonds of HNTs via hydrogen bonding interactions. Liu and coworkers prepared chitosan/HNTs bionanocomposite films via solution casting method. HNTs significantly improved the Young’s modulus and tensile strength of chitosan film with the loading of HNTs up to 7.5%. The reinforcing effect of HNTs on chitosan could be attributed to the dispersion state of nanoparticles in the matrix and their interfacial interactions [34]. Recently, Lisuzzo and coworkers introduced HNTs to Mater-Bi-based polymer matrix to fabricate biocompatible packaging materials and concluded that the HNTs improved the mechanical properties of the films. Compared to the pure biopolymer, the strongest enhancement of the tensile properties was achieved for Mater-Bi containing 10 wt% halloysite [10].

To explain the increase in the compressive elastic modulus of ternary HNTm/PHAD-Hgs and -Cgs in response to the increasing amount of HNT given in Figure 6, it is necessary to evaluate the change of the effective crosslink density $\nu_{e}$ in the hybrid network according to the amount of HNTs added to the ternary structure. Assuming the validity of the classical formulation of rubber elasticity to the ternary hybrid hydrogels, the effective crosslink density distribution through the ternary hybrid network was determined from the crosslinked terpolymer concentration in the as-prepared state and the compressive elastic moduli data *G*_0_ by:(1)$$\nu_{e}=\frac{G_{0}}{\beta RT\nu_{2}^{0}}=\frac{1}{NV_{1}}$$
where *N* is the average network chain length in the network; *V*_1_ is the molar volume of the polymerization solvent; and the factor β equals 1 for affine network structure and 1 − 2/*f_ip_* for a phantom network, in which *f_ip_* is the functionality of the inorganic particles which can be determined using the concentration (C*_ip_*, *w/v* %) and molecular weight (M_w_, g/mol) of the particles according to $f_{ip}=2\nu_{e}\nu_{2}^{0}M_{w}{/C}_{ip}$. Although it has not been proven by SEM images, since the ternary hybrid cryogels are assumed to have a porous structure, only the data of ternary hybrid hydrogels were used in the calculations. Figure 7 shows the variation of the structural network parameters ($\nu_{e}$ and *N*) of the ternary hybrid hydrogels as a function of HNTs content. Since the swelling capacity of the synthesized hybrid gels is not very high, the affine network model was used in the calculations. As shown by the open symbols in Figure 7, the effective crosslink density of ternary hybrid gels increased very rapidly with the addition of a small amount of HNT. Increasing HNT content resulted in an increase in the total number of crosslinked terpolymer chains, affecting the compressive elastic moduli of both HNTm/PHAD-Hgs and -Cgs for the fixed concentrations of the HEMA, AMPS, and DEAEMA in the terpolymer network. By nonlinear curve fittinh to the calculated $\nu_{e}$ values plotted against HNT content in the feed C_HNT_ % (*w/v*), one may get the following useful empirical equation that can help to determine the value of $\nu_{e}$ for this ternary network structure depending on the amount of HNT:(2)$$\nu_{e}=1.322+6.216C_{\mathrm{HNT}}-3.450{\left(C_{\mathrm{HNT}}\right)}^{2}+0.587{\left(C_{\mathrm{HNT}}\right)}^{3}$$
which indicates that the effective crosslink density for the ternary hybrid system can be given by a cubic polynomial as a function of HNT concentration. Figure 7 also shows the change in the average network chain length according to HNT concentration as HNT is added to the ternary network structure. It is clear that the sharp increase in the effective crosslink density causes the chain lengths in the network to decrease as HNT is added to the ternary network. The interchange of the calculated $\nu_{e}$ and *N* values as HNT particles are added to the ternary structure indicated that these particles, which are fixed in the matrix, act as additional crosslinking points in a larger terpolymer–nanotube interface because of the high aspect ratio (length/diameter) (10–50) of HNTs [35]. According to Liu et al. [36], few siloxane and hydroxyl groups are placed on the surface of HNTs, revealing that the tube–tube interactions are relatively weak due to chemical and geometrical aspects. Geometrically, the tube-like morphology of HNT with a proper aspect ratio creates few opportunities for large-area contact between the nanotubes, and HNTs do not tend to agglomerate. The mechanism of interaction of ternary polymer PHAD and HNTs occurs through hydrogen bonds between the oxygen atoms present in the nanotube structure and the protons of the amine and hydroxyl group of the terpolymer. The use of inorganic HNTs was found by Alhuthali and Low [37] to improve the mechanical performance of vinyl-ester nanocomposites. It was reported that the load transfer between matrix and reinforcing particles is efficient and ultimately results in better strength properties for the composites. Kim and coworkers [38] used HNT and biodegradable polylactic acid to fabricate nanocomposites by a melt-blending method with HNT compositions varying from 1 to 9 wt%. A decrease in tensile strength was observed for HNT contents exceeding 5 wt%. The elongation at break increased up to 5 wt% HNT content and then decreased with further increase of HNT.

### 2.6. Swelling Characteristics of Ternary Hybrid Gels

#### 2.6.1. Effect of HNTs on Swelling Characteristics in Water

To be comparable, the swelling characteristics of ternary hybrid hydrogels and cryogels were evaluated in deionized water at room temperature in comparison to the as-prepared state. The results in terms of the volume degree of swelling $\varphi_{V}$ are collected in Figure 8 as a function of HNT concentration. The variation of the physico-chemical network parameters—the terpolymer volume fractions of as-prepared state $\nu_{2}^{0}$ and swollen state $\nu_{2}^{}$ determined using Equations (4) and (5) is also presented as a function of HNT concentration in Figure 8. It can be seen that the experimental $\nu_{2}^{0}$ values calculated from the gel weights were compatible with the theoretical values. Optical images of swollen HNTm/PHAD-Hgs and -Cgs containing 0.30%, 2.20%, and 2.5% (*w/v*) of HNT are also presented for comparison. While the ternary hybrid hydrogels showed the highest swelling, cryogelation further reduced the degree of swelling in the ternary hybrid cryogels containing high amount of HNT. For both HNTm/PHAD-Hgs and -Cgs, the preparation of ternary hybrid gels by addition of various amount of HNT decreased the volume degree of swelling in comparison to HNT-free terpolymer gels. As the amount of HNT was increased up to 0.70%, there was a rapid decrease in the swelling degree of the ternary hybrid gels, and a 2.3-fold decrease in the swelling degree was observed compared to HNT-free gels. Then, they had a nearly constant value between 0.70% and 1.40% (*w/v*) HNT. The volume degree of swelling for 3.20% (*w/v*) HNT containing hydrogels and cryogels, HNT11/PHAD-Hgs and -Cgs, were the smallest of all samples.

The hydrophilicity of ternary hybrid gels prepared in this study is assigned to the presence of hydrophilic functional groups such as hydroxyl, carboxyl, sulfonic acid, and tertiary amine groups in their networks. Introducing HNTs to the ternary hybrid PHAD network increases the formation of hydrogen bonds between the SiO_2_ and the network components. For this reason, free water molecules do not interact as strongly with ternary hybrid networks as with HNT-free PHAD networks. The decrease in the swelling degree of the prepared hybrid gels is also supported by the increase in the effective crosslink density as the amount of HNT increases (Figure 7). For all HNT concentrations, the volume swelling degree of ternary hybrid cryogels was less than those of ternary hybrid hydrogels due to their porous structure. Sengel et al. successfully prepared super porous composite cryogels using carboxymethyl cellulose and HNT via cryogelation method [39] and examined the morphology of these gels in detail to show the effect of HNT on the porous structure. The effect of HNTs on the swelling behavior of HNT-containing nanocomposites reported by other researchers and the presented results for ternary hybrid HNTm/PHAD gels are consistent with their results. In their work, Luo and Mills studied the swelling behavior of chitosan/HNT nanocomposite hydrogels with the concentration of HNTs ranging from 1% to 5% [14]. The interfacial binding between HNTs and chitosan was achieved by electrostatic interactions and hydrogen bonding. The addition of HNTs increased the crosslink density and, thus, significantly reduced the swelling ratio in phosphate buffer saline. Sadegh-Hassani and Nafchi [40] introduced HNTs as the reinforcing agent to potato starch matrix to prepare bio-nanocomposite films. The HNTs are likely to bond with hydroxyl groups and other possible hydrogen or van der Walls bonds of starch macromolecules. The incorporation of HNTs (1%, 2%, 3%, and 5% of total starch solid) suppressed the diffusion of water into the structure, decreased the solubility in water from 35% to 23%. The water absorption capacity of the films significantly decreased by the increased extent of hydrogen bonds formed between HNTs and the matrix components.

#### 2.6.2. Effect of Temperature on Swelling Characteristics in Water

Figure 9 illustrates the effect of the swelling temperature as well as HNTs content on the extent of swelling of ternary hybrid HNTm/PHAD-Hgs and -Cgs in deionized water. Since the network in terpolymer structure contains 10 mol% DEAEMA monomer that is sensitive to temperature, it is expected that ternary hybrid HNTm/PHAD-Hgs and -Cgs exhibit temperature sensitive swelling behavior in water. Although the DEAEMA ratio in the structure is low, it was observed that the ternary hybrid gels exhibit temperature sensitive swelling behavior in the 25–75 °C temperature range. An important result in Figure 9 is that the cryogelation process did not alter the temperature sensitive swelling behavior of the ternary hybrid HNTm/PHAD-cryogels. The effect observed in ternary hybrid hydrogels was also observed in ternary hybrid cryogels. However, it was observed that the overall swelling rate of ternary hybrid cryogels is higher than ternary hybrid hydrogels. This can be explained by the porous structure of the ternary hybrid cryogels, although it cannot be supported by SEM images. The swelling ratio decreases as the temperature increases while the gels are in the swollen state at low temperatures, up to about 45 °C, since C=O groups (proton donors) and dialkylaminoethyl groups (proton acceptors) of PDEAEMA tend to form intermolecular hydrogen bonds with surrounding water. It was observed that both DEAEMA and HEMA monomers in ternary hybrid structure are dominant in the temperature-sensitive swelling behavior. While DEAEMA groups were dominant in the swelling behavior at low and moderate temperatures, it was clearly observed that HEMA groups were effective at high temperatures. Although the temperature-sensitive swelling tendency was the same, it was observed that the ternary hybrid hydrogels shrank more as the temperature increased, but the change in swelling profile of the ternary hybrid cryogels was not as significant. While the change in the swelling ratio due to the temperature variation is consistent with the characteristics of DEAEMA and HEMA monomer, it was observed that the swelling ratio of the ternary hybrid gels decreased as the amount of HNT increased, but it did not affect the temperature-sensitive swelling characteristics. In the 45–55 °C temperature range, the ternary hybrid gels shrunk due to increasing hydrophobic interactions between the pendant diethylaminoethyl groups and overwhelmed hydrogen bonding. According to Figure 9, the ternary hybrid gels gain a hydrophobic character, and their lower critical solution temperature (LCST) value is around 45 °C. However, it was observed that the ternary hybrid gels started to swell again interestingly in the range of 55–75 °C and exhibited an upper critical solution temperature feature, which can be interpreted as the formation of new hydrogen bonds during mixing with water. The hydroxyl groups in PHEMA are thus more bonded via hydrogen bond than the diethylaminoethyl groups in PDEAEMA at higher swelling temperatures. Despite having a fixed composition, the temperature dependence of the swelling changed from negative to positive as the swelling temperature increased. The positive swelling tendency of HEMA-based gels at high temperatures was first reported by Dusek and Janacek for copolymer gels of HEMA with methacrylamide (MA) [41]. They stated that, by increasing the amount of MA in the copolymer structure, the temperature dependence of swelling at 25–50 °C changed from negative to positive. The addition of more amide groups may increase in their role played in the formation of hydrogen bonds and weaken the extent of hydrophobic interactions. In a study investigating the phase separation properties of poly(dialkylaminoethyl methacrylate) polymers varying the substituents at the amino group (dimethylaminoethyl, diethylaminoethyl, and diisopropylaminoethyl), it has been reported that the polarity of the alkyl substituents and the carbon atoms attached to the aminoethyl group direct the order of transition, and the phase separation of PDEAEMA is mainly affected by less polar dialkylaminoethyl groups and backbone/carbonyl interactions [42]. The larger is the dialkylaminoethyl group, the lower is the temperature required to observe the LCST behavior.

#### 2.6.3. pH-Responsive Swelling Characteristics

The DEAEMA monomer in the terpolymer PHAD structure has pH-responsive characteristics due to the presence of tertiary amine groups in its structure. Therefore, it is expected that the prepared ternary hybrid gels have an increased hydrophilicity at pH values near and below the p*K*_a_ of PDEAEMA and should exhibit a pH-responsive swelling behavior to a certain extent. The pH-sensitivity of PDEAEMA has been used for a number of applications including targeting anticancer drug control release, stabilizer for emulsions and dispersions, and CO_2_-sensitive micellization [43,44,45,46]. pH-responsive swelling response of HNTm/PHAD-Hgs and -Cgs with various HNTs content was investigated due to the presence of pH-responsive DEAEMA moieties in the terpolymer network structure. The results are compared in Figure 10. As can be seen, the ternary hybrid gels were in a swollen state at low pH values with a high swelling, while the swelling ratio was substantially decreased with an increase in the swelling pH. In low pH conditions, tertiary amine side-groups in the PDEAEMA chains are protonated; hence, the ternary hybrid network has a transient cationic charge. The SiO_2_-based outer surface of HNTs is negatively charged in a wide pH range (2–8), enabling electrostatic attractions with cationic polymers, while the HNT lumen composed of Al_2_O_3_ presents a positive charge. Due to the electrostatic repulsion between cationic quaternary amine groups, HNTm/PHAD-Hgs and -Cgs tend to swell at low pH values and exhibit higher swelling capacity by protonating in the acidic pH region compared to the basic pH region. By deprotonating in acidic pH, which occurs at pH higher than 6.7, the electrostatic repulsion gradually disappears, and the gels tend to shrink; therefore, the swelling ratio is decreased substantially.

While the changes in the ternary hybrid gel volume and the swelling due to pH variations were consistent with the characteristics of the DEAEMA monomer, it was observed that, as the amount of HNT increased, the swelling rate of the hybrid gels, at all pH values, decreased in accordance with the swelling values in water. It was observed that the pH-responsive swelling tendency of the hybrid cryogels was the same as that of hybrid hydrogels. However, low-HNT hybrid cryogels had higher swelling capacity in the acidic region than the hybrid hydrogels. This can be explained by the fact that the hybrid cryogel structure is porous because it was observed that, as the pH value of the external solution increased, hybrid hydrogels shrank more, but the decrease in the swelling of hybrid cryogels was not so significant. In addition, with the increase of HNTs concentration, the degree of swelling decreased. The pH value at which the swelling decreased the most and the ternary hybrid gel passed from fully swollen to collapsed state was found to be compatible with the p*K*_a_ value of the PDEAEMA polymer (p*K*_a_ ≈ 6.7). This result is similar to those achieved by Marek and coworkers for DEAEMA and poly(ethylene glycol)-monomethyl ether monomethacrylate nanogels on the order of 200 nm using a free radical thermal inverse emulsion polymerization. The cationic nanogels synthesized with poly(ethylene glycol) tethers exhibited pH dependent swelling properties and had a transition from collapsed to swollen state at pH~7.2. The results obtained in this study show that molecules such as functionalized enzymes and glucose oxidase can be successfully incorporated into these systems [47].

#### 2.6.4. Salt-Induced Swelling Characteristics

Salt-induced swelling behavior (salting-out and salting-in effects) of ternary hybrid gels were examined in various KX solutions (X = halogen). The effect of different halide ions with a common cation (K^+^) on the swelling ratio of ternary HNTm/PHAD-Hgs and -Cgs is illustrated in Figure 11, Figure 12 and Figure 13 as a function of the concentration in aqueous KI, KBr, and KCl solutions. The results indicate that the $\varphi_{V}$ curves show differences for these three potassium salt solutions. While the ternary hybrid gels behave similarly in KBr and KCl solutions, their swelling tendency in KI solutions was different. Ternary hybrid gels exhibited “salting-in’’ behavior in KI salt solution, while a “salting-out’’ behavior was observed in KBr and KCl salt solutions. Comparing the swelling ratio of the ternary hybrid cryogels in the KI solution, it was found that the swelling ratio of low-HNT hybrid cryogels is higher than that of hybrid hydrogels. This result, which is parallel to pH- and temperature-sensitive swelling results, can also be associated with the porous structure of ternary hybrid cryogels.

The swelling ratio of ternary hybrid gels in the presence of I^−^ ions increased with increasing salt concentration (salting-in) (Figure 11), while the swelling ratio in the presence of Cl^−^ and Br^−^ ions decreased with increasing salt concentration (salting-out) (Figure 12 and Figure 13). As the concentration of KBr and KCl solutions increased (Figure 12 and Figure 13), the osmotic pressure difference between the ternary hybrid networks and the external solution decreased, resulting in reduced swelling of the ternary gels and shrinkage of the network structure by the effect of salt on the splitting of hydrogen bonds. It seems that the Br- and Cl- anions could specifically interact with the hydroxyl and amine groups, thus building physical bonds between them. In the presence of I^−^ ions, the swelling ratio changes slightly in the dilute KI solution and then increase at higher ionic concentration (Figure 11). When the swelling behavior of ternary hybrid cryogels and hydrogels in KBr and KCl solutions were compared, it was found that the swelling capacity of hybrid cryogels containing low HNT content was higher than that of hybrid hydrogels, but, when the amount of HNT was increased in the hybrid cryogel matrix, the swelling decreased compared to hybrid hydrogels. When the amount of HNT is low, the hybrid cryogel structure is more porous, but, when the amount of HNT is increased, as the polymer density in the unfrozen region increases more as a result of the cryoconcentration process, the swelling decreases. The swelling ratio of the ternary hybrid gels indicated an increase in the order of: Cl^−^ < Br^−^ < I^−^ for KCl, KBr, and KI, respectively. Therefore, the high swelling capacity of the ternary hybrid HNTm/PHAD-gels in KI solutions can be explained by the small charge/radius ratio of anion easily bound to tertiary amine group of DEAEMA. The larger anion tends to expand the network chains as it easily infiltrates the ternary cross-linked network. When the salt ions approach, the ion with the smaller charge density is easily polarized and tends to bind to the tertiary amine groups. Because it has the largest ionic radius, the negative charge of I^−^ makes the network swell more than that of Cl^−^. These observations agree with previously reported results for HEMA/N,N-dimethyl-(acrylamido propyl) ammonium propane sulfonate (HEMA/DMAAPS) copolymeric gels by Lee and Chen [48]. In the saline solution, the swelling ratios of HEMA/DMAAPS copolymer gels increased rapidly with increasing concentration of the salt, as well as with smaller ratio of charge/radius (I^−^, Br^−^, ClO^−4^, and NO^−3^). Conversely, when the salts possess a large charge density, the swelling ratios decrease with rise in salt concentration (F^−^ and CH_3_COO^−^).

## 3. Conclusions

A poly(alkyl methacrylate)-based ternary hybrid system consisting of halloysite nanotubes and tertiary amine functional groups in the form of cryogels and hydrogels was designed in an efficient and convenient way via freezing-induced gelation. The hydrophilicity and thermal and mechanical behaviors of ternary hybrid gels were investigated with different weight percents of HNTs. The effect of HNTs on the thermal stability of terpolymer matrix was evaluated by thermogravimetric analysis, and dispersion of HNTs in the terpolymer matrix was analyzed by XRD. The nature of interactions between ternary polymer and halloysite nanotubes in the hybrid matrix was investigated by FTIR spectroscopy. The mechanical performances of hybrid gels were improved by the addition of a small amount of HNTs, which indicated a homogenous distribution in the ternary matrix. The stronger are the host–terpolymer matrix interactions, the more important is the reinforcement effect as a result of the strong interfacial interactions and hydrogen bonding interactions. As HNT was added to the ternary network structure, a sharp increase in the effective crosslink density caused the chain lengths in the network to decrease. The addition of HNTs significantly reduced the swelling ratio of ternary hybrid hydrogels and cryogels. Despite having a fixed composition, ternary hybrid gels showed both negative and positive temperature-responsive swelling behavior depending on the swelling temperature. DEAEMA groups were dominant in swelling behavior at low and moderate temperatures as a result of hydrophobic–hydrophilic interactions, and the rising temperature led to a decrease in the swelling ratio. However, HEMA groups were effective at higher temperatures: the extent of hydrophobic interactions decreased and the role played by the hydrogen bonds increased, and, therefore, the ternary hybrid gels started to swell again. Besides, a significant decrease was observed in pH-responsive swelling capacity of ternary hybrid gels by incorporation of a small amount of HNT. Due to the electrostatic repulsion between terpolymer chains carrying quaternary amine groups, the ternary hybrid gels swelled at low pH conditions and shrank with increasing pH of the solution. To examine the effect of different halide ions (I^−^, Br^−^, and Cl^−^) with a common cation (K^+^), the salt-induced swelling of ternary hybrid gels was evaluated in aqueous salt solutions. Ternary hybrid gels showed “salting-out” behavior in the presence of Cl^−^ and Br^−^ ions, while “salting-in” behavior was observed in KI solutions of high concentrations. The results reveal that, depending on the application requirements, one can tune the hydrophilicity and overall elasticity of poly(alkyl methacrylate)-based terpolymers with respect to different loadings of HNTs. These results can provide a new perspective in the field of HNT-containing weak polybasic hybrid system design by demonstrating the importance of functional monomer selection and terpolymer composition.

## 4. Materials and Methods

### 4.1. Materials

Hydroxyethyl methacrylate (HEMA), 2-acrylamido-2-methyl-1-propanesulfonic acid (AMPS), and 2-(diethylamino)ethyl methacrylate (DEAEMA) were provided by Merck. The chemical crosslinker diethyleneglycol dimethacrylate (DEGDMA, Fluka), the redox-initiator system ammonium persulfate (APS, Merck, Darmstadt, Germany), and N,N,N′,N′-tetramethylethylenediamine (TEMED, Merck) were used as received. Halloysite nanotube (HNT) was supplied by ESAN Eczacibasi, Turkey. The elemental composition of HNT was (wt.%): SiO_2_, 43.30; Al_2_O_3_, 38.40; Fe_2_O_3_, 0.80; TiO_2_, 0.10. Disodium hydrogen phosphate (Merck), potassium dihydrogen phosphate (Riedel-de Haen, Darmstadt, Germany), hydrochloric acid (Merck), and potassium phosphate (J.T. Baker, Phillipsburg, New Jersey, USA) were used for pH-responsive swelling experiments. Potassium iodide (KI, Carlo Erba, Sabadell, Barcelona), potassium bromide (KBr, Merck), and potassium chloride (KCl, Merck) were used for the salt-responsive swelling experiments. Deionized water was used for all the synthesis stages as well as for swelling measurements.

### 4.2. Uniaxial Compressive Testing

The compressive modulus of ternary hybrid gels in as-prepared state and after equilibrium state were determined by measuring the deformation via uniaxial compressive testing. Upon completion of the polymerization reactions, the cylindrical gel specimens with initial diameter ≈4.0 mm were cut in pieces of 1.0 cm length for as-prepared state measurements. However, for swollen-state measurements, the size of the specimens was different based on their swelling capacities in water. Ternary hybrid specimens were uniaxially compressed between the fixed lower plate of apparatus and a vertically fitted circular probe with a diameter of 40 mm, each parallel to the other. Before the compression, initial undeformed diameter, length, and mass of all samples were measured, and, after adjusting the testing conditions, they were carefully placed on the fixed lower plate. A compressive load was transmitted vertically to the ternary hybrid specimen through load transducer with a fitted circular probe at a speed of 10 mm/min. After each compression, to follow the decrease in the length ΔL of the ternary hybrid gel, a digital comparator (IDC type Digimatic Indicator 543–262, Mitutoyo), which is sensitive to displacements of 10^−3^ mm, was used. The compressive force applied to the specimen (*f*) after each loading was calculated from the corresponding change in the mass *m*, as *f* = *mg*, using gravitational acceleration, *g* = 9.803 m/s^2^. The resulting deformation after each loading was recorded after 20 s of relaxation of chains by using load and displacement transducers. Using the undeformed length $L_{in}$ and deformed length L along the vertical axis, the variation in the length of the specimen after each loading was calculated as: $\Delta L=L_{in}-L$. It was then used for determination of the deformation ratio of specimen α, which can be calculated as the ratio of displacement ΔL to the undeformed length $L_{0}$ along the vertical axis. The compressive modulus *G* was determined from the slope of the linear dependence [49,50]:(3)$$\sigma =f/A=G(\alpha -\alpha^{-2})$$
where σ is the compressive stress in Nm^−2^ determined by the ratio of applied compressive force after each loading *f* to *A*, which is the area of the undeformed gel specimen calculated using the diameter as: $A=\pi {\left(D_{0}/2\right)}^{2}$. Considering the complexity of testing conditions, each compression was completed in less than 3 min to avoid the loss of water during the swollen-state measurements. For both as-prepared and swollen-state measurements, four ternary hybrid specimens were tested for each HNT concentration to get the mean value of the compressive strength and compressive modulus.

### 4.3. Physicochemical Characterization

ATR-FTIR was used to determine the functional groups of HNT-free and ternary hybrid gels. The spectra were recorded using with a Perkin Elmer pectrum 100 FTIR spectrometer by the ATR technique with a range of 4000–650 cm^−1^ at a resolution of 4 cm^−1^. The interlayer distances and the changes in nanotubular structure and HNT dispersed within the ternary hybrid network were studied by X-ray diffraction in a Bruker D8 Advance X-ray diffractometer (Cu-Ka tube, 40 kV, 40 mA); the range of the diffraction angle (2*θ*) was 4–50° at a scanning speed of 1° min^−1^. TGA analyses of HNT-free and ternary hybrid gels were carried out using a SEIKO EXSTAR 6200 Model TG/DTA instrument. The powdered samples were placed in an aluminum capsule and heated from 25 to 700 °C at a heating rate of 10 °C min^−1^ under a nitrogen atmosphere at a rate 150 mL min^−1^. Differential scanning calorimetry (DSC) measurements to determine the glass-transition temperature (T_g_) were made with a Perkin Elmer DSC4000 calorimeter with an accuracy of ±0.001. For the T_g_ measurements, 10 mg of hybrid samples dried under ambient conditions were put in an aluminum pan, equilibrated at −20 °C under a nitrogen flow rate of 20 mL min^−1^; heated from −20 to 200 °C at 10 °C/min to remove the thermal history; held at 200 °C for 10 min; cooled back from 200 to −20 °C at 10 °C/min; held for 10 min; heated again from −20 to 200 °C at 5 °C/min; and finally cooled back at the same rate. T_g_ was determined from the midpoint of the second heating curve, which was slow enough to detect the corresponding temperature.

### 4.4. Swelling Properties and Network Characteristics

The water absorption capacities of the HNT-free and ternary hybrid gels were determined by following their volume change after immersing in deionized water at room temperature. The swelling behavior in water was also investigated by changing the swelling temperature between 25 and 75 °C to test the temperature-responsive swelling behavior. Apart from the aquatic environment, the swelling characteristics of the gels were examined in pH-buffer solutions between pH 2.1 and 10.7. To investigate the effect of different halide ions with a common cation (K^+^) on the swelling degree, swelling measurements were conducted as a function of the salt concentration for KCl, KBr, and KI solutions ranging between 1.0 and 10^−5^ M. Upon measuring their initial diameter *D*_0_, which was ≈4 mm (equal to the diameter of the plastic syringe), the samples were immersed in water or the solution for at least two weeks until the thermodynamic equilibrium, and then the diameters of the swollen samples *D_eq_* were measured to calculate the equilibrium volume swelling capacity $\varphi_{V}$ from the differences between the diameters of the swollen and as-prepared samples as [51,52]:(4)$$\varphi_{V}=\frac{1}{\nu_{2}}=\frac{{\left(D_{eq}/D_{0}\right)}^{3}}{\nu_{2}^{0}}$$
where $\nu_{2}$ is the crosslinked terpolymer volume fraction in the swollen state and $\nu_{2}^{0}$ is that of in the relaxed state (just after polymerization). For the ternary hybrid gels, the experimental values of $\nu_{2}^{0}$ were evaluated using the following expression:(5)$$\nu_{2,\exp}^{0}={\left[1+\frac{\left(\left(m_{0}/m_{\mathrm{dry}}\right)-1\right)\rho }{d_{1}}\right]}^{-1}$$
where *m*_0_ and *m*_dry_ are the weights of the gel samples in the as-prepared state and after drying; *d*_1_ is the density of water, 1.0 g/mL; and *ρ* is the density of ternary PHAD gel, considered as *ρ* = 1.418 ± 0.035 g/mL. The theoretical values of $\nu_{2}^{0}$ were determined from the synthesis conditions as: $\nu_{2,theo}^{0}=10^{-3}C_{0}{\bar{V}}_{r}$, where C_0_ is the total molar concentration of the monomers and $\;{\bar{V}}_{r}$ is the average molar volume of ternary PHAD repeat units (in mL/mol). The experimental $\nu_{2}^{0}$ values given in Figure 9 are the mean of the measurements from the samples taken from four different syringes and the standard deviations were less than 3%. The experimental $\nu_{2}^{0}$ values and the gel fraction $w_{gel}$ values calculated from the gel weights are presented in Table 1 against the amount of HNT in the ternary network.

## Data Availability

The data presented in this study are available on request from the corresponding author. The data are not publicly available as they also forms part of an ongoing study.

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
