# Peer review of "One-Shot Preparation of Polybasic Ternary Hybrid Cryogels Consisting of Halloysite Nanotubes and Tertiary Amine Functional Groups: An Efficient and Convenient Way by Freezing-Induced Gelation"

_gels, 2021, doi:10.3390/gels7010016_

Round 1
Reviewer 1 Report
This paper reports about the preparation and characterization of hybrid organic/inorganic hydrogels. The work is carefully performed and the effect of several parameters (% of HNT, pH, T, ionic strength, type of electrolyte etc.) on the physicochemical properties of the hybrid gels. However, the description of the hydrogel preparation and presentation of the result must be improved. After the following points are addressed the paper can be accepted for publication in Gels
Introduction. The aim of the work is not clearly stated. We have to arrive to line 103 before the purpose of the paper is presented.
Line 57. The sentence “Compared to kaolin…” should be rewritten.
Line 80. “Subzero conditions”?
Line 105. “Responsive” to what
Line 112-118. It is not clear if “the freezing induced gelation” hydrogels (indicated later as Cgs) were also chemically crosslinked with DEGDMA. If not Cgs would be physical gels whereas those indicated with Hgs are chemical gels. This point should be clearly stated, in the introduction and in in scheme 1.
TGA results.
Figure 2, table 2 and text. In the figure 2 on the y scale it is reported “Mass Loss (%)” but actually it is not. It is the M(T)/M0 %, where M(T) is the mass at the temperature T and M0 is the mass at RT. A legend should be added to figure 2A to identify the dotted lines.
Line 240. (Å) ?
Line 322. “dissociative amorphous” ?
Figure 5. Smaller symbols for the data point should be used.
Figure 7. What is the meaning of Naffine (the label of the ordinate axis)?
Line 634. The proposed explanation for the different effect of I- compared to Cl- and Br- on swelling is not clear.
Figures 9-12. For some of these figures it would be better to show φV on a linear scale instead of on a logarithmic scale.
Author Response
Manuscript ID: gels-1095388
Type of manuscript: Article
Title: One-shot preparation of polybasic ternary hybrid cryogels consisting
of Halloysite nanotubes and tertiary amine functional groups: Efficient and
convenient way by freezing-induced gelation
Authors: Nur Sena Okten Besli, Nermin Orakdogen *
Gels
01/30/2021
Authors' response to reviewers' comments, list of changes and point-to-point answers
Reviewer #1:
This paper reports about the preparation and characterization of hybrid organic/inorganic hydrogels. The work is carefully performed and the effect of several parameters (% of HNT, pH, T, ionic strength, type of electrolyte etc.) on the physicochemical properties of the hybrid gels. However, the description of the hydrogel preparation and presentation of the result must be improved. After the following points are addressed the paper can be accepted for publication in Gels
Introduction. The aim of the work is not clearly stated. We have to arrive to line 103 before the purpose of the paper is presented.
Response: We thank the reviewer for this suggestion. An explanation indicating the purpose of the study are added to the first paragraph of the introduction, line 41-45.
Line 57. The sentence “Compared to kaolin…” should be rewritten.
Response: We corrected the sentence “Compared to kaolin, by a layer of water molecules, the unit layers are separated from eachother” on page 2, line 60-61 in green as:
“Compared to kaolin, the unit layers are separated by a layer of water molecules”.
Line 80. “Subzero conditions”?
Response: “Subzero conditions” are mentioned in which the polymerization is carried out below the freezing temperature of the polymerization solvent. Therefore, the relevant sentence was corrected and rewritten on page 2, line 80-83 in green as follows:
“Various approaches to improve the mechanical weakness of nano (hybrid) materials have been reported in the literature [17-20]; among them, the most common procedure is to carry out the polymerization at a temperature below the freezing point of polymerization solvent [21-23].”
Line 105. “Responsive” to what
Response:In this sentence, the word “responsive” means sensitivity to pH and temperature. The sentence was rewritten on page 3, line 105-108 in green as follows:
“The purpose of this research is to develop a synthetic way to prepare nanostructured ter-nary systems with typical pH and temperature-sensitive alkylmethacrylate-based poly-mer properties, but with enhanced flexibility and thermal properties of HNTs depending on the network topology.”
Line 112-118. It is not clear if “the freezing induced gelation” hydrogels (indicated later as Cgs) were also chemically crosslinked with DEGDMA. If not Cgs would be physical gels whereas those indicated with Hgs are chemical gels. This point should be clearly stated, in the introduction and in in scheme 1.
Response:The crosslinker DEGDMA monomer was used in the synthesis of all hydrogels and cryogels prepared within the scope of this study. Therefore, all hybrid gels should be considered as chemically crosslinked. To prevent misunderstanding on this subject, necessary explanatory sentences have been added in the introduction, on page 3, line 114-118 as follows: “The development of one-shot preparation of polybasic ternary hybrid cryogels containing nanoparticle HNT, neutral monomer 2-hydroxyethyl methacrylate (HEMA), anionic monomer 2-acrylamido-2-methyl-1-propane sulfonic acid (AMPS), cationic monomer (DEAEMA) in the presence of chemical crosslinker diethyleneglycol dimethacrylate (DEGDMA) is presented.”
We also revised Scheme 1 and the legend of Scheme 1 on page 5:
“Scheme 1. Schematic diagram illustrating the structure and preparation process of ternary hy-brid HNTm/PHADgels in the presence of chemical crosslinker diethyleneglycol dimethacry-late (DEGDMA).”
TGA results.
Figure 2, table 2 and text. In the figure 2 on the y scale it is reported “Mass Loss (%)” but actually it is not. It is the M(T)/M0 %, where M(T) is the mass at the temperature T and M0 is the mass at RT. A legend should be added to figure 2A to identify the dotted lines.
Response: As suggested, we revised Figure 2 on page 7 and reported the y scale as M(T)/M0 %. A legend has been added to figure 2 to identify the dotted lines. Table 2 and corresponding text on page 7, line 239-245 have been also revised in green:
To determine the thermal stability of terpolymer hybrid materials as well as their compositional properties, Figure 2(A) shows the plot of M(T)/M0 %, the fraction of the sample mass remaining after heating, where M0 refers to initial mass; M(T) refers to the mass at temperature T. The derivative weight loss (% / oC) curves of pristine HNT, HNT-free hydrogel and cryogel (HNT0/PHAD-Hgs and Cgs), ternary hybrid hydrogel and cryogel (HNT10/PHAD and HNT11/PHAD-Hgs and Cgs) samples prepared at HNT con-tent 2.80 %(w/v) and 3.20 %(w/v) are shown in Fig.2(B).
Figure 2. (A) M(T)/M0 % and (B) differential thermogravimetric (DTG) curves of HNT-free hydrogel and cryogel (HNT0/PHAD-Hgs and Cgs), ternary hybrid hydrogel and cryogel (HNT10/PHAD and HNT11/PHAD-Hgs and Cgs) samples prepared at HNT content 2.80 %(w/v) and 3.20 %(w/v). (C) M(T)/M0 % and DTG curve of pristine HNT.
Line 240. (Å) ?
Response: Here, the term “Å” refers angstrom, Ao. We corrected the notation on page 7, line 247. From XRD results presented on page 10, we found that the Halloysite which is used in the synthesis is HNT-10 Ao.
Introduction, line 50-52: Naturally-occurring HNT belongs to dioctahedral 1:1 clay mineral of the kaolin group with a predominantly hollow tubular structure with chemical formula Al2Si2O5(OH)4·nH2O, where n = 0 for halloysite-(7 Ao) and n = 2 for halloysite-(10 Ao), respectively.
Page 9, line 317-321: The XRD pattern of pristine HNT reflected a monoclinic-domatic crystallographic character of Halloysite-10 Å. The characteristic sharp peak at 8.86° (2q) of HNT corresponding to d001 basal spacing of 9.966 Ǻ, determined by using Bragg's Law occurred due to HNT multiwall reflection. This result indicated that HNT was mainly in the hydrated form and typically referred as 10Ǻ-Halloysite.
Line 322. “dissociative amorphous” ?
Response: We revised the corresponding sentence on page 10, line 331-333 in green as follows:
“The XRD pattern of HNT-free hydrogel and cryogel (HNT0 / PHAD-Hgs and Cgs) showed a wide reflection centered ~ 17.42 ° (2q), (d = 5.083 Ao) which was attributed to the amorphous structure of terpolymer PHAD.”
Figure 5. Smaller symbols for the data point should be used.
Response: On page 12, Figure 5 has been revised using smaller symbols.
Figure 7. What is the meaning of Naffine (the label of the ordinate axis)?
Response: Here, Naff is the average network chain length calculated assuming affine network model. On page 14, line 435-436, we added an explanation as:
“Since the swelling capacity of the synthesized hybrid gels is not very high, the affine net-work model has been used in the calculations.”
We also revised the legend of Figure 7 on page 15 as follows:
Figure 7. Variation of the average network chain length calculated assuming affine network model and the effective crosslink density of ternary hybrid HNTm/PHAD-Hgs network as a function of HNT content.
Line 634. The proposed explanation for the different effect of I- compared to Cl- and Br- on swelling is not clear.
Response: The swelling ratio of the ternary hybrid gels indicated an increase in the order of Cl- < Br- < I- for KCl, KBr and KI, respectively. The following explanation has been included on page 21 line 678-684 as follows:
Therefore, the high swelling capacity of the ternary hybrid HNTm / PHAD-gels in KI solu-tions can be explained by the small charge / radius ratio of anion easily bound to tertiary amine group of DEAEMA. The larger anion tends to expand the network chains as it easi-ly infiltrates the ternary cross-linked network. When the salt ions approach, the ion with the smaller charge density is easily polarized and tend to bind to the tertiary amine groups. Because it has the largest ionic radius, the negative charge of I- makes the network swell larger than that of Cl-.
Figures 9-12. For some of these figures it would be better to show φV on a linear scale instead of on a logarithmic scale.
Response: Figure 9 has been revised and φV has been presented on a linear scale instead of on a logarithmic scale. Regarding the representation of Figures 11 and 12, these figures were revised with the recommendation of the 2nd reviewer and the related figure were redrawn in two dimensions.
Reviewer 2 Report
This manuscript titled " One-shot preparation of polybasic ternary hybrid cryogels consisting of Halloysite nanotubes and tertiary amine functional groups: Efficient and convenient way by freezing-induced gelation" describes the fabrication of HNT-ternary hybrid hydrogel via conventional polymerization and cryo-polymerization. The gels were characterized by FTIR and XRD to determine the molecular structure and were evaluated for its compression strength and swelling ratio under different temperature, pH and salt concentration. In general, the experiments are well-designed and the discussion is sufficient back by data and previous reports.
Below are my comments to the authors
- One major deficiency observed in this manuscript is the lack of elaboration or emphasis on the result of the cryo-induced gels. With the title giving much emphasis on cryo-induced and cryogel, readers are expecting more detail explanation on the difference in the result caused by cryo-induced, especially the sections on the effect of temperature, pH and salt concentration on swelling characteristic.
- The compression result of non-swollen cryogel in figure 5 is missing.
- I suggest to conduct SEM for the gels to demonstrate their porous structure.
- I suggest change the way of presenting the result in figure 9, 11, 12, as the difference in the data cannot be observed in 3D graphs.
Author Response
Manuscript ID: gels-1095388
Type of manuscript: Article
Title: One-shot preparation of polybasic ternary hybrid cryogels consisting
of Halloysite nanotubes and tertiary amine functional groups: Efficient and
convenient way by freezing-induced gelation
Authors: Nur Sena Okten Besli, Nermin Orakdogen *
Gels
01/30/2021
Authors' response to reviewers' comments, list of changes and point-to-point answers
Reviewer #2:
This manuscript titled " One-shot preparation of polybasic ternary hybrid cryogels consisting of Halloysite nanotubes and tertiary amine functional groups: Efficient and convenient way by freezing-induced gelation" describes the fabrication of HNT-ternary hybrid hydrogel via conventional polymerization and cryo-polymerization. The gels were characterized by FTIR and XRD to determine the molecular structure and were evaluated for its compression strength and swelling ratio under different temperature, pH and salt concentration. In general, the experiments are well-designed and the discussion is sufficient back by data and previous reports.
Below are my comments to the authors
- One major deficiency observed in this manuscript is the lack of elaboration or emphasis on the result of the cryo-induced gels. With the title giving much emphasis on cryo-induced and cryogel, readers are expecting more detail explanation on the difference in the result caused by cryo-induced, especially the sections on the effect of temperature, pH and salt concentration on swelling characteristic.
Response: We thank the reviewer for this suggestion. With the recommendation of the reviewer, we supported the results for cryogels by discussing the effect of temperature, pH, and salt type&concentration on swelling property.
To discuss the effect of cryogelation on swelling characteristics of ternary hybrid hydrogels and cryogels in water, an explanation has been added on page 15, line 485-487, and page 17, line 508-510.
For temperature-dependent swelling results, an explanation has been added on page 17 line 540-546 and page 18, line 551-556.
To support the pH-responsive swelling results, an explanation has been included to compare pH-dependent swelling behavior results on page 20, line 617-623.
More explanation and comparison has been added to discuss the salt-sensitive swelling behavior of ternary hybrid cryogels and hydrogels on page 21 line 650-654 and line 670-676.
- The compression result of non-swollen cryogel in figure 5 is missing.
Response: In this study, for hybrid cryogels; only the swollen moduli of the synthesized cryogels were measured and the modulus in the post-synthesis state was not measured. Since the cryogels are synthesized at -18 °C, there are micro-macro frozen regions in the three-dimensional netwok structure in the post-synthesis state and technically, the modulus is not measured in the post-synthesis situation since these regions will act to increase the modulus. For this reason, no changes have been made in Figure 5.
- I suggest to conduct SEM for the gels to demonstrate their porous structure.
Response: Yes, the reviewer is right. However, since we did not have the opportunity to provide SEM images of terpolymer nanocomposite cryogels in the current period, we could not add the images.
- I suggest change the way of presenting the result in figure 9, 11, 12, as the difference in the data cannot be observed in 3D graphs.
Response: Figure 9 has been revised and φV has been presented on a linear scale instead of on a logarithmic scale as there is a suggestion from the 1st reviewer regarding this issue. Figures 11 and 12 were redrawn in two dimensions. Since the swelling capacities of gels are close to each other, there are overlapping data in two-dimensional representation. For this reason, we think that a 3D representation would be more appropriate.